

# The status and influencing factors of abnormal fetal pregnancy outcomes in 265 cases in China: a retrospective study

Jing Ruan[1,*], Xuemei Zhong[2,*], Jiaxuan Mai[3], Cuifen Liu[4] and Huiyang Ding[3]

[1] Department of Nursing, Guangdong Women and Children Hospital, Guangzhou, Guangdong Province, China
[2] Breast Surgery, Guangdong Women and Children Hospital, Guangzhou, Guangdong Province, China
[3] Neonatal Surgery Department, Guangdong Women and Children Hospital, Guangzhou, Guangdong Province, China
[4] Fetal Life Cycle Clinic, Guangdong Women and Children Hospital, Guangzhou, Guangdong Province, China
* These authors contributed equally to this work.

Corresponding author
Jing Ruan, gdsfyrj@163.com

## ABSTRACT

**Background:** With the advancement of prenatal diagnosis technology, the detection rate of fetal abnormalities continues to increase, imposing a significant burden on both society and families. A retrospective analysis of essential information about pregnant women, such as their pregnancy history and delivery details, is crucial for understanding the primary factors that influence pregnancy outcomes in women with fetal abnormalities. This analysis is of great significance for improving the level of pregnancy management and outcomes in pregnant women with fetal abnormalities.

**Objective:** To retrospectively analyze the pregnancy outcomes of women with fetal abnormalities and explore the factors that influence these outcomes.

**Methods:** Pregnant women's pregnancy outcomes were collected from the medical information system and through telephone follow-ups. The chi-square test and logistic regression were used to analyze the factors influencing pregnancy outcomes.

**Results:** Among 265 pregnant women diagnosed with fetal abnormalities, 190 chose to continue the pregnancy, while 75 chose to terminate it. Pregnant women with multiple fetal abnormalities (OR = 3.774, 95% CI [1.640–8.683]) were more likely to choose termination of pregnancy (TOP), and pregnant women who were advised to terminate their pregnancy or make a careful choice were more likely to terminate the pregnancy (OR = 41.113, 95% CI [11.028–153.267]).

**Conclusion:** The number of organs involved in fetal abnormalities and treatment recommendations were identified as the primary factors influencing pregnancy outcomes. Improving awareness of maternal health care during pregnancy, early pregnancy screening technology, and a multidisciplinary diagnosis and treatment approach are of great significance in assisting pregnant women in making informed decisions and improving fetal prognosis.

## INTRODUCTION

Fetal abnormalities are the leading cause of early abortion, stillbirth, late infant death, and congenital malformations (*Brent, 2004*). According to the World Health Organization (WHO), congenital anomalies have an incidence of 2–3% and account for 240,000 newborn deaths worldwide every year (*WHO, 2023*). Congenital anomalies not only seriously affect the survival and quality of life of children but also cause great pain and heavy economic burden on families and society (*Boyd et al., 2008*; *Heaney, Tomlinson & Aventin, 2022*). Owing to environmental pollution, deficiencies in maternal nutrition, radiation exposure, poor lifestyle habits, and advanced maternal age, China has a high incidence of birth defects (*Chen et al., 2022*; *Ghazi et al., 2021*; *Tan, Sethi & Sulaiman, 2022*; *Zhang et al., 2020*) of approximately 5.6%, equivalent to approximately 900,000 new birth defects every year (*Li & Di, 2021*).

Progress in prenatal diagnosis technology and disease research has enabled the detection and treatment of an increasing number of fetal abnormalities during the early stages of development (*Haxel et al., 2022*). However, the decision to continue or terminate a pregnancy with fetal abnormalities is influenced by various factors, including national laws and regulations, religious beliefs, ethics, socioeconomic conditions, education level, and severity of fetal abnormalities (*Hjort-Pedersen et al., 2022*; *Kerns et al., 2012*; *Pusayapaibul, Manonai & Tangshewinsirikul, 2022*). In the UK, approximately 37% of pregnant women choose to terminate their pregnancy when diagnosed with fetal abnormalities (*Heaney, Tomlinson & Aventin, 2022*). In the United States, this number ranges from 47 to 90% (*Kerns et al., 2012*), and in Thailand, more than half of pregnant women opt for termination (*Pusayapaibul, Manonai & Tangshewinsirikul, 2022*). In China, when a fetal abnormality is diagnosed, the pregnant woman and her family members are informed about the type, treatment, and prognosis of the diagnosed abnormal condition. For a confirmed fatal fetal abnormality, termination of pregnancy (TOP) is recommended with informed consent from the pregnant woman and her family members. For non-fatal fetal abnormalities with a high risk of treatment and poor prognosis, doctors will inform the woman and her family about the perinatal prognosis and provide advice for careful consideration. In such cases, the woman and her family may choose to terminate the pregnancy, in which they would sign an informed consent form; alternatively, they may choose postpartum treatment, in which case, close follow-up after birth is recommended. For treatable fetal abnormalities with a good prognosis, doctors may recommend continuing the pregnancy. However, the decision to continue pregnancy ultimately depends on the pregnant woman. The rate of abnormal fetal termination of pregnancy is 70.66%, in rural areas, termination rates have exceeded 90% (*Xie et al., 2020*).

Guangdong Women and Children Hospital, located in one of the economically developed regions of China, in order to enhance the management of fetal birth defects throughout the entire pregnancy process, starting from May 2022, the hospital has integrated various departments such as prenatal diagnosis, obstetrics, neonatology, neonatal surgery, pediatric surgery, neurological rehabilitation, pediatric genetics and metabolism, stomatology, ENT, ophthalmology, reproduction, ultrasound diagnosis, and

radiology (MRI), established a multidisciplinary "Fetal Life Cycle Clinic" to provide comprehensive treatment for pregnant women diagnosed with fetal abnormalities.

The purpose of this study was to retrospectively analyze the clinical data and pregnancy outcomes of women with abnormal fetal pregnancies, analyze the factors affecting the outcomes of abnormal fetal pregnancies, and provide a reference for prognosis consultation and treatment of fetal dysplasia.

## MATERIALS AND METHODS

### Patients and data selection

Data were obtained from the medical information system and through telephone follow-up of pregnant women who visited the "Fetal Life Cycle Clinic" at Guangdong Women and Children Hospital from May 2022 to March 2023. Data on maternal age, diagnostic gestational age, fetal anomaly type (single or multiple abnormalities), treatment recommendation, number of births, history of abortion, history of fetal abnormalities, mode of conception, complication of pregnancy, degree of education, employment, and pregnancy outcomes were collected. Pregnant women were included if they were diagnosed with fetal abnormalities by any of the following methods: fetal nuchal translucency value measurement, B-ultrasound, magnetic resonance imaging, non-invasive DNA testing, fluorescence *in situ* hybridization, amniocentesis, or genetic testing. Pregnant women were excluded if they declined follow-up or lacked complete information.

There are 13 major types of fetal abnormalities: abnormalities of the central nervous system, cardiovascular system, respiratory system, urinary system, digestive system, musculoskeletal system, anterior abdominal wall, facial abnormalities, fetal tumors, fetal edema, reproductive system, genetic/chromosomal abnormalities, and viral infections. According to the number of organs involved, fetal abnormalities are classified as single or multiple abnormalities (*Li & Luo, 2018*).

Pregnancy outcomes included delivery and termination of pregnancy (TOP) (induced labor or fetal reduction). In this study, labor was defined as a woman's decision to carry the pregnancy to its completion and deliver the fetus. Induction of labor is defined as the voluntary decision of pregnant women and their families to undergo therapeutic TOP after receiving genetic counseling and providing informed consent. Fetal reduction is defined as the termination of fetal development due to dysplasia, malformation, or multiple pregnancies to ensure normal survival and development of a healthy fetus. In this study, both induction of labor and fetal reduction were classified as termination of pregnancy (TOP).

### Statistical analysis

Descriptive analysis was used to assess the general characteristics of the pregnant women with fetal abnormalities. Quantitative data conforming to a normal distribution were expressed as mean ± standard deviation, while categorical data were presented as constituent ratio or rate (%), and the chi-square test or univariate analysis was used for comparison between groups. Variables with statistical significance in the chi-square test or

single-factor analysis were selected as the initial tiers of the candidate variables. Considering that in single-factor analysis, differences between the results could not truly reflect the effect of this factor on the outcome event, logistic regression was adopted to adjust and correct confounding factors; $P < 0.05$ was considered statistically significant.

## Ethical considerations

This study was approved by the Ethics Committee of Guangdong Women and Children's Hospital (approval number: 202301125). Data were obtained through a review of outpatient case records and telephone follow-up. Considering that the patients came from all over the country, it was difficult for them to sign the informed consent form face-to-face. Therefore, to save on travel costs and time for patients, we utilized a pre-made article informed consent form during the telephone follow-up and explained the background, purpose, and significance of the study along with their rights and interests in the study. Subsequently, the patients could decide orally whether to participate in the study and withdraw from the study at any time.

# RESULTS

## Characteristics of pregnant women for diagnosing fetal abnormalities

A total of 398 pregnant women visited our hospital between May 2022 and March 2023. Among them, 273 had a birth outcome (delivery or termination of pregnancy). However, three participants did not answer the phone during the telephone follow-up and five refused to participate in the study. Ultimately, 265 pregnant women were included in this study. A total of 190 pregnant women gave birth either through natural labor ($n = 91$) or cesarean section ($n = 99$); however, six newborns died within 1 month of birth. Of the remaining 75 pregnant women with fetal abnormalities, 67 chose to induce labor, and eight pregnant women with twin pregnancies opted for fetal reduction. In addition, the average age of the pregnant women was $30.05 \pm 4.60$ years (range, 20–47 years), with 257 singleton pregnancies and eight twin pregnancies. The gestational age at diagnosis of fetal abnormalities was $\leq 13^{+6}$ weeks in 5 case (1.9%), $14$–$27^{+6}$ weeks in 183 case (69.1%), and $\geq 28$ weeks in 77 cases (29%). 36 pregnant women were advised to carefully consider whether to continue with the termination, and 229 pregnant women were advised to continue with pregnancy follow-up (Table 1).

In our study, there were 211 cases of single fetal abnormalities and 54 cases of multiple fetal abnormalities. For improved statistical classification, all instances of abnormal fetal diagnoses were calculated based on their frequency and occurrence rate. A total of 330 cases with fetal abnormalities were diagnosed, including 60 cases (18.2%) with central nervous system abnormalities, 52 cases (15.8%) with cardiovascular system abnormalities, 53 cases (16.10%) with respiratory system dysplasia, 54 cases (16.4%) with urinary system dysplasia, 37 cases (11.2%) with digestive system dysplasia, 23 cases (7.0%) with musculoskeletal system dysplasia, 19 cases (5.8%) with facial dysplasia, and six cases (1.8%) with gene/chromosome abnormalities. There were 16 cases (4.8%) of fetal tumors, three cases (0.9%) of anterior abdominal wall dysplasia, three cases (0.9%) of fetal hydrops, two cases (0.6%) of reproductive system dysplasia, and two cases (0.6%) of viral infection

**Table 1 Comparison of pregnancy outcomes in pregnant women with different characteristics of fetal abnormalities.** By comparing the general basic characteristics of pregnant women with abnormal fetuses, the variables with statistical differences were preliminarily identified.

| Variable | Number of cases (n = 265) | Pregnancy outcome | | $x^2$ | P |
|---|---|---|---|---|---|
| | | Delivery (n = 190) | Termination of pregnancy (n = 75) | | |
| **Age (y) (%)** | | | | 0.524 | 0.914 |
| 18–24 | 26 (9.8) | 20 (7.5) | 6 (2.3) | | |
| 25–29 | 101 (38.1) | 72 (27.2) | 29 (10.9) | | |
| 30–34 | 97 (36.6) | 68 (25.7) | 29 (10.9) | | |
| ≥35 | 41 (15.5) | 30 (11.3) | 11 (4.2) | | |
| **Diagnostic gestational age (W) (%)** | | | | 10.843 | 0.004 |
| ≤13$^{+6}$ | 5 (1.9) | 2 (0.8) | 3 (1.1) | | |
| 14–27$^{+6}$ | 183 (69.1) | 123 (46.4) | 60 (22.6) | | |
| ≥28 | 77 (29) | 65 (24.5) | 12 (4.5) | | |
| **Fetal anomaly type (%)** | | | | 24.825 | <0.001 |
| Fetal single abnormality | 211 (78.9) | 166 (62.6) | 45 (17.0) | | |
| Fetal multiple abnormalities | 54 (21.1) | 24 (9.1) | 30 (11.3) | | |
| **Treatment recommendation (%)** | | | | 82.430 | <0.001 |
| Recommended to terminate the pregnancy or choose carefully | 36 (13.6) | 3 (1.1) | 33 (12.5) | | |
| Recommended continuation of pregnancy | 229 (86.4) | 187 (70.6) | 42 (15.8) | | |
| **Number of births (%)** | | | | 0.008 | 0.927 |
| Unipara | 93 (35.1) | 67 (25.3) | 26 (9.8) | | |
| Pluripara | 172 (64.9) | 123 (46.4) | 49 (18.5) | | |
| **History of abortion (%)** | | | | 0.098 | 0.754 |
| Yes | 110 (41.5) | 80 (30.2) | 30 (11.3) | | |
| No | 155 (58.5) | 110 (41.5) | 45 (17.0) | | |
| **History of fetal abnormalities (%)** | | | | 0.352 | 0.553 |
| Yes | 18 (6.8) | 14 (5.3) | 4 (1.5) | | |
| No | 247 (93.2) | 176 (66.4) | 71 (26.8) | | |
| **Mode of conception (%)** | | | | 0.410 | 0.522 |
| Natural conception | 259 (97.7) | 185 (69.8) | 74 (27.9) | | |
| Artificial impregnation | 6 (2.3) | 5 (1.9) | 1 (0.4) | | |
| **Complication of pregnancy (%)** | | | | 3.269 | 0.071 |
| Yes | 28 (10.6) | 16 (6.0) | 12 (4.5) | | |
| No | 237 (89.4) | 174 (65.7) | 63 (23.8) | | |
| **Degree of education (%)** | | | | 5.001 | 0.172 |
| Primary school and below | 3 (1.1) | 3 (1.1) | 0 (0.0) | | |
| Junior high school | 64 (24.2) | 41 (15.5) | 23 (8.7) | | |
| Education technical secondary school or above | 77 (29.1) | 54 (20.4) | 23 (8.7) | | |
| Specific colleague course or above | 121 (45.6) | 92 (34.7) | 29 (10.9) | | |
| **Employment (%)** | | | | 0.023 | 0.880 |
| Be in employment | 150 (56.6) | 107 (40.4) | 43 (16.2) | | |
| Unemployed | 115 (43.4) | 83 (31.3) | 32 (12.1) | | |

(Continued)

| Variable | Number of cases (n = 265) | Pregnancy outcome | | $x^2$ | P |
|---|---|---|---|---|---|
| | | Delivery (n = 190) | Termination of pregnancy (n = 75) | | |
| **Place of abode (%)** | | | | 0.194 | 0.660 |
| Guangzhou | 90 (34) | 63 (23.8) | 27 (10.2) | | |
| Outside Guangzhou | 175 (66) | 127 (47.9) | 48 (18.1) | | |

(Fig. 1). The most common fetal abnormalities diagnosed in pregnant women include abnormalities in the central nervous, urinary, cardiovascular, and respiratory systems. The two primary abnormalities that resulted in TOP were also related to central nervous and cardiovascular system abnormalities (Fig. 2).

### Influencing factors of fetal abnormal termination of pregnancy

Chi-square test analysis showed significant differences in the gestational age at diagnosis of fetal abnormalities, the number of fetal abnormalities, diagnostic gestational age, and treatment recommendations of doctors among pregnant women ($P < 0.05$). However, there were no significant differences in maternal age, number of births, abortion history, history of fetal abnormalities, mode of conception, pregnancy complications, education level, employment status, or place of abode (Table 1).

We conducted binary logistic regression analysis using gestational age (first, second, or third trimester), number of organs affected by fetal abnormalities (single abnormality/ multiple abnormalities), and treatment recommendations. The gestational age at diagnosis, number of fetal abnormalities, and diagnosis and treatment recommendations of doctors showed statistically significant differences in pregnancy outcomes. Based on the literature (*Crowe et al., 2018*; *Michalik & Preis, 2014*) and clinical experience, maternal age, number of births, abortion history, history of fetal abnormalities, mode of conception, pregnancy complications, and educational level may be potential confounding factors affecting pregnancy outcomes. Therefore, to enhance the accuracy of the statistical results, we adjusted for the confounding factors. Consequently, we discovered that diagnostic gestational age did not have a significant impact on pregnancy outcomes ($P > 0.05$). The treatment recommendations of doctors, as well as the number of fetal abnormalities, significantly affected pregnancy outcomes in pregnant women ($P < 0.05$).

Pregnant women with multiple fetal abnormalities were 3.774 times more likely to undergo labor induction than those with a single fetal abnormality (OR = 3.774, 95% CI [1.640–8.683]). Pregnant women who were advised to terminate their pregnancies or carefully consider whether to continue gestation were 41.113 times more likely to undergo labor induction compared to those who were advised to continue the pregnancy (OR = 41.113, 95% CI [11.028–153.267]), (Table 2).

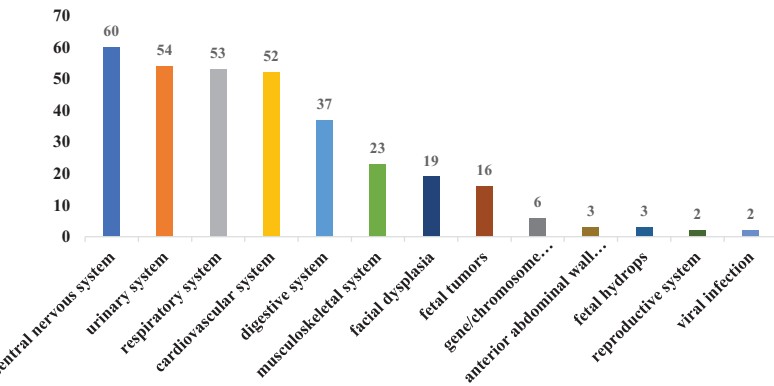

**Figure 1  Classification of fetal abnormalities.**

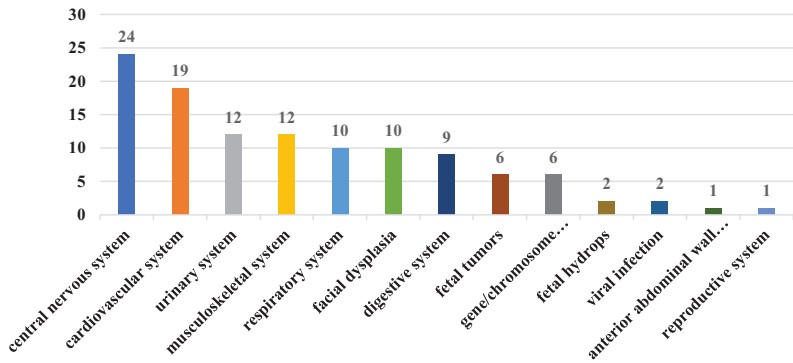

**Figure 2  Classification of fetal abnormalities with termination of pregnancy.**

**Table 2  Analysis of influencing factors of pregnancy outcome in pregnant women with fetal abnormalities.** Through the adjustment of confounding factors, the factors related to pregnancy outcomes were identified.

| Factors | Unadjusted | | | Adjusted | | |
|---|---|---|---|---|---|---|
| **Treatment recommendation** | OR | 95% CI | P | OR | 95% CI | P |
| Recommended continuation of pregnancy* | | | | | | |
| Recommended to terminate the pregnancy or choose carefully | 33.982 | [9.754–118.392] | <0.01 | 41.113 | [11.028–153.267] | <0.01 |
| **Fetal anomaly type** | | | | | | |
| Fetal single abnormality* | | | | | | |
| Fetal multiple abnormalities | 3.154 | [1.461–6.811] | 0.003 | 3.774 | [1.640–8.683] | 0.002 |
| **Diagnostic gestational age** | | | | | | |
| ≥28 W* | | | | | | |
| 14–27$^{+6}$ W | 2.717 | [1.145–6.447] | 0.023 | 2.454 | [0.988–6.095] | 0.053 |
| 13$^{+6}$ W | 7.302 | [0.858–62.170] | 0.069 | 7.709 | [0.776–76.545] | 0.081 |

**Notes:**
The variables adjusted for were gestational age, number of births, history of abortion, history of fetal abnormalities, mode of conception, complication of pregnancy, degree of education.
* Means reference.

## DISCUSSION

This study included 265 pregnant women diagnosed with fetal abnormalities. Of these, 1.9% were in the first trimester, 69.1% in the second trimester, and 29% in the third trimester. The second trimester is the optimal period for detecting fetal abnormalities (*Edwards & Hui, 2018*; *Rydberg & Tunón, 2017*). With advancements in prenatal screening technology, it is now possible to detect fetal chromosomal abnormalities by measuring the thickness of the translucency in the fetal neck and analyzing serum markers during early pregnancy. In addition, prenatal ultrasound can also detect abnormalities in other areas, including the neck (92%), limbs (34%), face (34%), genitourinary organs (34%), and heart (53%). Therefore, an increasing number of fetal abnormalities can be detected in the first trimester. In our study, fetal abnormalities were found in only 1.9% of the pregnant women during the first trimester of pregnancy. This low diagnostic rate may be attributed to the poor compliance of pregnant women during the first trimester of pregnancy with regard to screening as well as variations in screening technology across different hospitals (*Bardi et al., 2022*; *Karim et al., 2017*). It is suggested that the publicity and health education on birth defects prevention and control be further strengthened in primary hospitals. This can enhance the efficient utilization of prenatal diagnosis techniques and improve the level of diagnosis, thereby reducing diagnostic errors such as missed diagnosis and misdiagnosis (*Sharp & Alfirevic, 2014*).

The fetal abnormalities detected in this study were most prevalent in the central nervous system, cardiovascular system, and respiratory system. Among them, The central nervous system had the largest proportion of abnormalities, accounting for 18.2% of all developmental abnormalities. It is also the most common abnormality observed in pregnant women undergoing labor induction (*Akinmoladun, 2021*). The central nervous system and cardiovascular system abnormalities are among the most common congenital abnormalities in China, accounting for one-third of all cases (*Dai et al., 2011*), From 2009 to 2011, the Chinese government invested 320 million yuan in a program aimed at providing free folic acid supplementation to pregnant women in rural areas, with the goal of preventing neural tube defects, and the effect was remarkable, as the nationwide incidence of neural tube defects decreased by 62.3% (*Liu et al., 2015*). However, neurological abnormalities remained the most common fetal abnormalities in our study, which may be attributed to the advancements in prenatal diagnosis technology in recent years, enabling the detection of an increasing number of neurological abnormalities at an early stage. Additionally, the increase in advanced maternal age and exposure to harmful substances in the living environment of women of childbearing age may contribute to this trend. Abnormalities in the central nervous system may also result in intellectual and physical abnormalities in newborns after birth, with a mortality rate of 36.45%, furthermore, 62.79% of the surviving children were found to have neurodevelopmental disorders (*Hart et al., 2022*; *Struksnæs, Blaas & Vogt, 2019*; *Tan, Sethi & Sulaiman, 2022*). Therefore, most families often choose to induce labor, considering the poor prognosis of the fetus and the impact on the quality of family life in the later stage.

We found that the diagnosis of multiple abnormalities was an important factor affecting termination of pregnancy (TOP), which was consistent with the findings of *Rauch et al. (2005)*. Multisite abnormalities are frequently linked to chromosomal abnormalities, which increase the likelihood of uncertain fetal prognosis. As a result, pregnant women and their families may be more likely to consider abortions.

In most cases, doctors' recommendations reflect the severity of the disease, which is another important factor that may affect pregnancy outcomes in women with fetal abnormalities, consistent with previous findings (*Hjort-Pedersen et al., 2022*; *Horn-Oudshoorn et al., 2023*). In our study, one fetus was diagnosed with spina bifida and the other with trisomy 18. Both conditions are associated with a high stillbirth rate. Even in cases of live birth, these conditions had a significant impact on the baby's later motor and intellectual development. Therefore, considering the poor prognosis, doctors recommend termination of pregnancy (TOP). The other 34 cases of fetal abnormalities diagnosed in pregnant women were non-fatal, but they required complex medical, surgical, or plastic treatment at a high cost and had potentially serious consequences. Doctors informed both parents about perinatal prognosis, enabling them to decide whether to proceed with the pregnancy or terminate it.

Previous studies (*Bardi et al., 2022*; *Crowe et al., 2018*) have shown that pregnant women diagnosed with fetal abnormalities in the first trimester were more likely to terminate their pregnancies. This was mainly because the fetus was not fully developed in the first trimester and chromosomal abnormalities and severe structural malformations could be screened at this stage. Even if pregnant women chose to terminate their pregnancies, their physical and psychological damage was lower than that in the second and third trimesters. However, no statistical difference was found in the diagnosis of gestational age in our study, which may be attributed to the small sample size included in the first trimester. *Brooks et al. (2019)* investigated the preference of 101 pregnant women for prenatal testing and termination of pregnancy and found that more than 60% of pregnant women reported that the reason for termination of pregnancy was mainly related to the severity of the disease (such as early infant death, severe intellectual disability, hemoglobinopathy, and amelia) and had no relationship with age, delivery time, abortion history, or previous history of fetal abnormalities. This result is consistent with our findings. *Schechtman et al. (2002)* found that the educational level of pregnant women is inversely proportional to the likelihood of terminating pregnancy. Pregnant women with a high school education were less likely to choose to terminate a pregnancy when faced with fetal abnormalities than those without a high school education. This trend may be attributed to education level. High acceptance of disease knowledge in communication with doctors is expected. However, this phenomenon was not observed in this study. This discrepancy may be attributed to the fact that 74.7% of the 265 pregnant women in our study had educational levels above high school. Consequently, the overall level of awareness was higher, and more emphasis was placed on the severity of fetal abnormalities.

In our study, 86.4% of pregnant women were advised to continue with the pregnancy, but the actual termination rate was only 15.8%, which was significantly lower than the rates reported in previous studies (*Boyd et al., 2008*). This could be attributed to advancements

in fetal and pediatric medicine, which have improved the prognosis of the disease. Additionally, the hospital implemented a multidisciplinary approach for diagnosis and treatment. In the past, prenatal consultation for fetal abnormalities was primarily conducted by obstetricians, and the assessment of prognosis and perinatal management was not as thorough. Therefore, TOP has become the primary treatment recommendation. At present, our hospital has established a multidisciplinary diagnosis and treatment center called the "Fetal Life Cycle Clinic." The clinic consists of various departments and practices, including prenatal diagnosis, obstetrics, neonatology, neonatal surgery, pediatric surgery, neurological rehabilitation, pediatric genetics and metabolism, oral medicine centers, otolaryngology departments, ophthalmology, reproductive centers, ultrasound diagnosis departments, and radiology departments (nuclear magnetic resonance). In this study, we found the significance of multidisciplinary diagnosis and treatment, a practice that can be replicated by other hospitals in the future. Integrating resources, leveraging the advantages of specialties, providing patients with comprehensive diagnosis and treatment plans, and managing the entire perinatal period can improve pregnancy outcomes by avoiding unnecessary labor induction (*Kim et al., 2019*; *Menzel et al., 2018*).

In conclusion, The outcome of pregnant women with fetal abnormalities is mainly related to the number of fetal abnormalities and the doctor's treatment recommendations. To reduce the occurrence of birth defects, it is essential to enhance primary prevention efforts, raise awareness about the importance of maternal healthcare during pregnancy, and improve the quality of prenatal diagnosis in medical facilities. For pregnant women diagnosed with fetal abnormalities, it is important to provide prenatal counseling, as well as social and professional support. This will help them and their families to make informed decisions regarding their pregnancies. Multidisciplinary diagnosis and treatment approaches can lead to timely and effective comprehensive diagnosis and treatment, improve fetal prognosis, and promote the development of disciplines.

Our study has some limitations. First, this was a single-center, retrospective study conducted at a hospital in China. Selection bias cannot be completely excluded, the sample size was small, and the conclusions could only be considered as hints and suggestions. Therefore, in future research, it is necessary to conduct prospective, multicenter, and large-sample studies. Second, most pregnant women did not undergo invasive prenatal diagnosis owing to economic constraints, which could potentially lessen the influence of chromosomal or genetic abnormalities on pregnancy outcomes. Third, patient privacy should be considered, important variables, such as family income, personal income level of pregnant women, religious beliefs, and psychological quality, were not captured by the information system. This may result in overestimation of the influence of other factors. Therefore, in future studies, health information regarding pregnancy and childbirth should be improved.

## ACKNOWLEDGEMENTS

We are especially grateful to the patients and all the authors who agreed to participate in this study.

### Funding

The authors received no funding for this work.

### Competing Interests

The authors declare that they have no competing interests.

### Author Contributions

- Jing Ruan conceived and designed the experiments, analyzed the data, prepared figures and/or tables, authored or reviewed drafts of the article, and approved the final draft.
- Xuemei Zhong conceived and designed the experiments, analyzed the data, prepared figures and/or tables, authored or reviewed drafts of the article, and approved the final draft.
- Jiaxuan Mai conceived and designed the experiments, performed the experiments, prepared figures and/or tables, and approved the final draft.
- Cuifen Liu conceived and designed the experiments, authored or reviewed drafts of the article, and approved the final draft.
- Huiyang Ding performed the experiments, prepared figures and/or tables, authored or reviewed drafts of the article, and approved the final draft.

### Human Ethics

The following information was supplied relating to ethical approvals (*i.e.*, approving body and any reference numbers):

Guangdong Women and Children Hospital (202301125).

### Data Availability

The raw data is available in the Supplemental File.

### Supplemental Information

Supplemental information for this article can be found online at http://dx.doi.org/10.7717/peerj.17284#supplemental-information.

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
