# Peer review of "The status and influencing factors of abnormal fetal pregnancy outcomes in 265 cases in China: a retrospective study"

_PeerJ, doi:10.7717/peerj.17284_

## Round 0.1 · original submission · Minor Revisions

In addition to the reviewer's comments, one of the limitations of a single-center study is generalizability; The authors should add this to their limitations.

Reviewer 1 ·

Basic reporting

Overall, the article is well written. The literature references are appropriate and up-to-date. The structure is clear. The source of data is clearly stated. However, there are some grammar mistakes and format issues to be improved. I also suggest adding marginal numbers to the first row (Number of
cases, Delivery, Termination of pregnancy) in Table 1.

Experimental design

The sample size is limited, especially, for the Diagnostic gestational age(W)(%) <= 13 subgroup. If possible, researchers should try to increase the sample size.

It is not clearly stated in the statistical method session whether the researchers control for any potential confounding variables in the logistic regression models. I suggest elaborating more on this part and explaining in more detail what variable we need to adjust and why. More rigorous statistical methods are needed.

Validity of the findings

The conclusion sentence 'The gestational age at diagnosis, the number of abnormal diagnoses, and treatment recommendations are factors that influence the decision to induce labor in pregnant women with fetal abnormalities.' is ambiguous and a little misleading. We can only conclude the association but not the causation using the method. The interpretation of statistical results should be improved.

Additional comments

No additional comments

Reviewer 2 ·

Basic reporting

The manuscript is well-structured, adhering to professional standards with clear, unambiguous English. The literature review provides an adequate background, setting a solid foundation for the study's relevance and necessity. Figures and tables are appropriately used to summarize the study's findings, enhancing the reader's comprehension. However, the manuscript could benefit from explicitly stating the availability and access details of the raw data, aligning with PeerJ's policy for transparency and reproducibility.

Experimental design

The study presents original primary research that fits within the journal's scope, addressing a significant gap in the literature regarding the outcomes of pregnancies affected by fetal abnormalities in a Chinese hospital context. The research question is clearly defined, relevant, and meaningful. The methods are described with enough detail to allow replication, and ethical considerations are adequately addressed. Nevertheless, this section could be enhanced by further discussing the selection of statistical methods and elaborating on how potential biases were managed.

Validity of the findings

The findings are supported by robust, statistically sound data and are well-aligned with the initial research question. The study's conclusions are appropriately limited to the evidence presented. However, the impact and novelty of the findings could be further emphasized, detailing how this research contributes to existing knowledge and suggesting avenues for meaningful replication. Clarifying how these findings could influence practice or policy would add value.

Additional comments

Raw Data Accessibility: Provide a clear statement regarding the availability of raw data, possibly through a public repository, to facilitate transparency and replication efforts.
Impact and Novelty: Enhance the discussion on the study's impact by comparing its findings with existing literature, highlighting its unique contributions to the field.
Encourage Replication: Outline potential settings or populations where replication could yield additional insights, promoting further research based on this study's findings.

---

## Round 0.2 · accepted · Accept

All comments have been adequately addressed

Reviewer 1 ·

Basic reporting

The authors carefully addressed the issues, the manuscript is in good shape now.

Experimental design

The authors carefully discuss the sample size issues.

Validity of the findings

The authors carefully discussed the statistical analysis and interpretation.

Additional comments

The authors carefully addressed the issues raised in the first round of review, the manuscript is in good shape now.

Reviewer 2 ·

Basic reporting

The authors have shared raw data and related documents which is quite helpful for future research kept consistent.

Experimental design

Here is a good improvement.

Validity of the findings

no comment

Additional comments

A minor issue in reference list should be corrected: the contents of the 3rd and 4th paper are almost identical (Bardi et al), except for the year 2022a and 2022b.